

# The influence of instrumental line shape degradation on NDACC gas retrievals

Youwen Sun[1, 3]+, Mathias Palm [2]+, Cheng Liu [3, 4, 1]*, Frank Hase [5], David Griffth [6],

Christine Weinzierl [2], Christof Petri [2], Wei Wang [1], and Justus Notholt [2]

(1 *Key Laboratory of Environmental Optics and Technology, Anhui Institute of Optics*

*and Fine Mechanics, Chinese Academy of Sciences, Hefei 230031, China*)

(2 *University of Bremen, Institute of Environmental Physics, P. O. Box 330440, 28334*

*Bremen, Germany*)

(3 *Center for Excellence in Urban Atmospheric Environment, Institute of Urban*

*Environment, Chinese Academy of Sciences, Xiamen 361021, China*)

(4 *University of Science and Technology of China, Hefei, 230026, China*)

(5 *Karlsruhe Institute of Technology (KIT), Institute for Meteorology and Climate*

*Research (IMK-ASF), Karlsruhe, Germany*)

(6 *School of Chemistry, University of Wollongong, Northfields Ave, Wollongong, NSW,*

*2522, Australia* )

+These two authors contributed equally to this work

**Abstract:**

Instrumental line shape (ILS) degradation can cause significant biases between global FTIR (Fourier transform infrared) networks if not properly treated. Currently, how ILS degradation influences the global NDACC (Network for Detection of Atmospheric Composition Change) gases retrieval and how much ILS deviation is acceptable for each NDACC gas are still not fully quantified. We simulated ILS degradations with respect to typical types of misalignment, and compared their influence on each NDACC gas. The sensitivities of total column, root mean square of fitting residual (RMS), total random uncertainty, total systematic uncertainty, total uncertainty, degrees of freedom for signal (DOFs), and profile with respect to different levels of ILS degradation for all current NDACC gases, i.e., $O_3$, $HNO_3$, HCl, HF, $ClONO_2$, $CH_4$, CO, $N_2O$, $C_2H_6$, and HCN, were investigated. The influence of an imperfect ILS on NDACC gases retrieval were assessed, and the consistency under different meteorological conditions and solar zenith angles (SZA) were examined.

Correspondence to: Cheng Liu (chliu81@ustc.edu.cn)



The study concluded that the influence of ILS degradation can be approximated by the
linear sum of individual modulation efficiency (ME) amplitude influence and phase
error (PE) influence. The PE influence is of secondary importance compared with the
ME amplitude influence. For total column retrieval, the stratospheric gases are more
sensitive to ILS degradation than the tropospheric gases. For profile retrieval, the
positive ME has more influence on tropospheric gases than the stratospheric gases. In
contrast, the negative ME has more influence on stratospheric gases than the
tropospheric gases. In order to suppress the influence on total column for $ClONO_2$ and
other NDACC gases within 10% and 1%, respectively, the permitted maximum ILS
degradation for each NDACC gas was deduced (summarized in Table 5).

**Key words:** NDACC, FTIR, Instrumental line shape, Profile retrieval
**1 Introduction**
In order to achieve consistent results between different FTIR (Fourier transform
infrared) sites, the TCCON (Total Carbon Column Observing Network,
http://www.tccon.caltech.edu/) and NDACC (Network for Detection of Atmospheric
Composition Change, http://www.ndacc.org/) have developed strict data acquisition
and retrieval methods to minimize site to site differences (Hase et al., 2012; Wunch et
al., 2010 and 2011; Washenfelder, 2006; Messerschmidt et al., 2010; Kurylo, 1991;
Davis et al., 2001; Schneider, et al.,2008; Kohlhepp et al., 2011; Hannigan et al., 2009;
Vigouroux et al., 2008 and 2015). Interferograms are acquired with similar
instruments operated with common detectors, acquisition electronics and/or optical
filters. These interferograms are first converted to spectra and then these spectra are
analyzed using dedicated processing algorithms, i.e., GFIT, PROFFIT or SFIT
(Wunch et al., 2010 and 2015; Hase et al., 2006; Hannigan and Coffey, 2009).
Typically, the TCCON network only uses the Bruker 125HR instruments
(http://www.tccon.caltech.edu/; https://www.bruker.com/) with specified settings
(entrance aperture, amplification of the detected signal). In the NDACC network,
other instruments are used as well, e.g., the Bruker M series, a BOMEM DA8 in



Toronto, Canada and a self-built spectrometer in Pasadena, USA
(http://www.ndacc.org/; https://www.bruker.com/). FTIR spectrometers are highly
precise and stable measurement devices and the instrumental line shapes (ILSs) not
far from the theoretical limit if carefully aligned. However, their alignment can
change abruptly as a consequence of operator intervention or drift slowly due to
mechanical degradation over time (Olsen et al., 2004; Duchatelet et al., 2010; Hase et
al., 2012; Feist et al., 2016). Moreover, the NDACC observation may change the
entrance field stop size if incident radiation changes. This practice may introduce a
dependency of the instrument alignment status on the optical settings because the
mechanical errors between different field stops may be non-negligible and
inconsistent (Sun et al., 2017). Biases between sites would arise if all these
misalignments are not properly characterized.
The TCCON network assumes an ideal ILS in spectra retrieval, and the maximum
ILS degradation is prescribed as 5% for the modulation efficiency (ME) amplitude
(Wunch et al., 2011 and 2015). This assumption still holds within the required
accuracy of the results. In the NDACC gases retrieval, the ILS can be assumed as
ideal if spectrometer is well aligned, or if misalignment exists, described by LINEFIT
results derived from dedicated cell measurements or retrieved together with the gas
profile from an atmospheric spectrum using a polynomial (Vigouroux et al., 2008 and
Vigouroux et al., 2015). How these ILS treatments influence the NDACC gases
retrieval and how much ILS deviation from unity is acceptable for each NDACC gas
if an ideal line shape is assumed are still not fully quantified, and it may be better to
assume an ideal ILS. The practice of co-retrieving ILS parameters from atmospheric
spectra without dedicated cell measurements is not to be recommended because the
observed shapes of spectral lines are exploited primarily for inferring the vertical
distribution of the trace gases, the ILS and the trace gas profiles have similar effects
on the line shape, i.e., changing the shape and width of the line. Overlapping lines, i.e.,
due to interfering gases may introduce an asymmetry in the absorption lines which
may be undistinguishable from an ILS phase deviation.
This paper investigates the influence of ILS degradation on NDACC gas





retrievals and deduces the maximum ILS deviations allowable for suppressing the
influence within a specified acceptable ranges.
**2 Characteristics of ideal and imperfect ILSs**
The ILS is the Fourier transform of the weighting applied to the interferogram.
This weighting consists of two parts: an artificially applied part to change the
calculated spectrum and an unavoidable part which is due to the fact that the
interferogram is finite in length (box car function), the divergence of the beam is
non-zero (due to the non-zero entrance aperture), and several other effects which are
due to misalignment (Davis et al., 2001, chapter 9). The ILS consisting of only the
unavoidable parts of the line shape is called the ideal line shape.
The theoretical ideal ILS as defined in equation (3), when the instrument is well
aligned, is a convolution of sinc and rectangular functions (defined in equations (1)
and (2)), representing the finite length of the interferogram and the finite circular field
of view (FOV) of the spectrometer (Davis et al., 2001).
$$SINC(\sigma, L) = 2L \frac{\sin(2\pi\sigma L)}{2\pi\sigma L} \tag{1}$$

$$RECT(\sigma, \sigma_0, \theta) = \begin{cases} \dfrac{2}{\sigma_0 \theta^2} & if -0.5\sigma_0\theta^2 \le \sigma \le 0 \\ 0 & otherwise \end{cases} \tag{2}$$

$$ILS(\sigma, \sigma_0, L, \theta) = SINC(\sigma, L) * RECT(\sigma, \sigma_0, \theta) \tag{3}$$

where $\sigma$ is the wavenumber, $\sigma_0$ is the central wavenumber, $L$ is the optical path
difference (OPD) and $\theta$ is the angular radius of the circular internal FOV of the
spectrometer. For standard NDACC measuring conditions, $L \ge 180$ cm and $\theta$ defined
by the entrance field stop size in the light path.
The LINEFIT software calculates the deviation of the measured ILS from the
ideal ILS (Hase et al., 2001 and 2012). It retrieves a complex ME as a function of
OPD, which is represented by a ME amplitude and a phase error (PE) (Hase et al.,
1999). The ME amplitude is connected to the width of the ILS while the PE quantifies
the degree of ILS asymmetry. For a perfectly aligned spectrometer, it would meet the





ideal nominal ILS characteristics if smear and vignetting effects were neglected, and
thus have an ME amplitude of unity and a PE of zero along the whole interferogram.
However, if a FTIR spectrometer is subject to misalignment, the ME amplitude would
deviate from unity and the PE deviate from zero (Hase et al., 2012). This results in an
imperfect ILS.

## 3 Simulation of ILS degradation

We use the program ALIGN60 to simulate ILS degradation in a high resolution
FTIR spectrometer typically used in the NDACC network. As a part of LINEFIT,
ALIGN60 is a package for simulation of the ILS of misaligned cube-corner
interferometers. It generates trustworthy results with respect to all types of
misalignment (Hase et al., 1999). In this simulation, the entrance beam section was
assumed to be circular with a diameter of 8.0 cm. The ILS was only calculated from
positive side of interferogram. The smear and vignetting effects were not taken into
account. The misalignment of a FTIR spectrometer can be expressed via two
perpendicular axes perpendicular to the beam direction. For a circular entrance beam,
the same misalignment in either direction results in a similar ILS. Thus, this work
only considers misalignment in one axis.
The misalignments as inputs of ALIGN60 are listed in Table 1 and the resulting
ILSs are shown in Fig. 1. All types of misalignment cause nonlinear ME deviations
except decentering of measuring laser ($c$) and the constant shear ($d$) which mainly
affect PE and result in linear PE deviation. Two types of ILS degradation are evident,
one is referred to as positive ME and has a ME amplitude of larger than unity. The
other one is referred to as negative ME and has a ME amplitude of less than unity.
Typically, the increasing misalignment ($b$, $f$, $h$ or $i$) causes negative ME amplitude and
the decreasing misalignment ($e$, $g$ or $j$) causes positive ME amplitude. For the same
misalignment amplitude, the decreasing misalignment causes more ME deviation than
the increasing misalignment. Regardless of positive or negative ME, the ME deviation
shape depends on misalignment type and the same misalignment amplitude causes the
same deviation in ME amplitude. The decentering of the entrance filed stop is





equivalent to the linear increasing misalignment.

## 4 NDACC gases retrieval

### 4.1 Retrieval strategy

The influence of ILS degradation on all current NDACC gases, i.e., $O_3$, $HNO_3$, HCl,
HF, $ClONO_2$, $CH_4$, CO, $N_2O$, $C_2H_6$, and HCN, is investigated here. Typical
atmospheric vertical profiles of these gases are shown in Fig.2. There are five
stratospheric gases and five tropospheric gases. The retrieval settings as recommended
by the NDACC for all these gases are listed in Table 2. The latest version of profile
retrieval algorithm SFIT4 v 0.9.4.4 is used (http://www.ndacc.org/). The basic
principle of SFIT4 is using an optimal estimation technique for fitting
calculated-to-observed spectra (Rodgers, 2000; Hannigan and Coffey, 2009). All
spectroscopic line parameters are adopted from HITRAN 2008 (Rothman et al., 2009)
in this study. This might not be ideal, but we keep it to achieve consistent results. A
priori profiles of pressure, temperature and water vapor for the measurement days are
interpolated from the National Centers for Environmental Protection and National
Center for Atmospheric Research (NCEP/NCAR) reanalysis (Kalnay et al., 1996). A
priori profiles of the target gases and the interfering gases except $H_2O$ use the
WACCM4 (Whole Atmosphere Community Climate Model) model data. We follow
the NDACC standard convention with respect to micro windows (MWs) selection and
the interfering gases consideration (https://www2.acom.ucar.edu/irwg/links). No
de-weighting signal to noise ratios (SNR) are used except for CO and HCl which
utilize a de-weighting SNR of 500 and 300, respectively.
The selection of the regularization (a priori covariance matrix $S_a$ and SNR) cannot
be easily standardised because it depends on the real variability for each gas. In
optimal estimation, the selection of $S_a$ is very important in the inversion process and,
together with the measurement noise error covariance matrix $S_\varepsilon$, will lead to the
following averaging kernel matrix $A$ (Rodgers, 2000):

$$A = G_y K_x = (K_x^T S_\varepsilon^{-1} K_x^T + S_a^{-1})^{-1} K_x^T S_\varepsilon^{-1} K_x \tag{4}$$

where $G_y$ is the sensitivity of the retrieval to the measurement. $K_x$ is weighting
function matrix or Jacobian matrix that links the measurement vector $y$ to the state
vector $x$ : $\Delta y = K_x \Delta x$. $A$ characterizes the vertical information contained in the FTIR
retrievals. In this study, we assume $S_\varepsilon$ to be diagonal and its diagonal elements are the





inverse square of the SNR. The vertical information content of the retrieved target gas
profile can be quantified by the number of degrees of freedom for signal (DOFs),
which is the trace of **A**, defined in Rodgers (2000) by:
$$d_s = tr(\mathbf{A}) = tr((\mathbf{K}_x^T \mathbf{S}_\varepsilon^{-1} \mathbf{K}_x^T + \mathbf{S}_a^{-1})^{-1} \mathbf{K}_x^T \mathbf{S}_\varepsilon^{-1} \mathbf{K}_x^T) \qquad (5)$$
The diagonal elements of $\mathbf{S}_a$ represent the assumed variability of the target gas
volume mixing ratio (VMR) at a given altitude, and the off diagonal elements
represent the correlation between the VMR at different altitudes. We can see in Table
3 that, except CO and HCN, the target gases are using an a priori covariance matrix
with diagonal elements constant with altitude corresponding to 10, 20, 50 or 100 %
variability; the largest variabilities are for $HNO_3$, HCl and $ClONO_2$. For CO, the
diagonal elements of $\mathbf{S}_a$ correspond to 27% from ground to 34 km and decrease down
to 11% at the top of atmosphere. For HCN, the diagonal elements of $\mathbf{S}_a$ correspond to
79% from ground to 5 km and decrease down to 21% at the top of atmosphere. No
correlation of off diagonal matrix elements is used in all retrievals except for $ClONO_2$
which uses exponential correlation with a HWHM (half with at half-maximum) of 8
km. The SNR values for all retrievals are the real values taken from each individual
spectrum. The ILSs for all retrievals are using the simulations in section 3.
**4.2 Averaging kernels**
Beside the a priori covariance matrix $\mathbf{S}_a$ and $\mathbf{S}_\varepsilon$, the averaging kernel matrix **A**
also depends on retrieval parameters including solar zenith angle (SZA), the spectral
resolution, and the choice of spectral micro windows (MW). The rows of **A** are the so
called averaging kernels and they represent the sensitivity of the retrieved profile to
the real profile. Their FWHM is a measure of the vertical resolution of the retrieval at
a given altitude. The area of averaging kernels represents sensitivity of the retrievals
to the measurement. This sensitivity at altitude $k$ is calculated as the sum of the
elements of the corresponding averaging kernels, $\sum_i A_{ki}$. It indicates the fraction of
the retrieval at each altitude that comes from the measurement rather than from the a
priori information (Rodgers, 2000). A value close to zero at a certain altitude indicates
that the retrieved profile at that altitude is nearly independent of measurement and is
therefore approaching the a priori profile.
The averaging kernel matrices of these ten NDACC gases are shown in Fig. 3.
Fig. 4 is the corresponding averaging kernels and their areas. The altitude ranges with



sensitivity larger than 0.5 and the corresponding DOFs are summarized in Table 3.
These sensitive ranges indicate that the retrieved profile information comes by more
than 50% from measurement, or, in other words, that the a priori information
influences the retrieval by less than 50%. Each gas has different sensitive range. The
sensitive range for HCN, CO and $C_2H_6$ is mainly tropospheric, and for $ClONO_2$, HCl
and HF is mainly stratospheric. $O_3$, $CH_4$ and $N_2O$ have high retrieval sensitivity in
both troposphere and stratosphere. The $HNO_3$ has high retrieval sensitivity in
stratosphere and in atmospheric boundary layer below 1.5 km.
**4.3 Error analysis**
As listed in Table 2, we classified errors as systematic or random according to
whether they are constant between consecutive measurements, or vary randomly. For
comparison, the error items considered in error analysis are the same for the retrieval
of all gases. The smoothing error $\mathbf{E}_s$ is calculated via equation (6), the measurement
error $\mathbf{E}_m$ is calculated via equation (7), and all other error items $\mathbf{E}_{var}$ are calculated via
equation (8) (Rodgers, 2000).
$$\mathbf{E}_s = (\mathbf{A} - \mathbf{I})\mathbf{S}_a(\mathbf{A} - \mathbf{I})^T \tag{6}$$

$$\mathbf{E}_m = \mathbf{G}_y\mathbf{S}_\varepsilon\mathbf{G}_y^T \tag{7}$$

$$\mathbf{E}_{var} = \mathbf{G}_y\mathbf{K}_{var}\mathbf{S}_{var}\mathbf{K}_{var}^T\mathbf{G}_y^T \tag{8}$$

where $\mathbf{S}_{var}$ is the error covariance matrix of *var*. $\mathbf{K}_{var}$ is weighting function matrix of
*var*. Here *var* refers to one of the error items in Table 2 except smoothing error and
measurement error. In this study, the a priori error covariances for all non-retrieval
parameters are set the same for all gases retrieval.
Figs.5 and 6 show the error components contributing to the systematic error and
random error covariance matrices of all NDACC gases, as well as the combined errors.
The structure in the error profiles shape reflects the effect of the propagation of
different errors in the retrieval process. The dominant sources of systematic errors and
random errors for all gases are listed in Table 4. For most gases, the dominant sources
of systematic errors are smoothing error, line intensity error and line pressure
broadening error. The dominant sources of random errors are measurement error and
zero level.
**5 ILS influence study**





This section presents the ILS influence study, whereby the degraded ILSs that
simulated by ALIGN60 are used in the SFIT forward model, and the fractional
difference (D%) in various quantities for each gas relative to the retrieval with an
ideal ILS are computed. For each gas, this section only selects one typical spectrum
for study. The consistency of the resulting deduction will be evaluated in section 6. All
spectra were recorded on a clear day at Hefei on February 16, 2017. For all spectra
used in this study, the actual ILS degradation of the FTIR spectrometer is less than
1.3% and can be regarded as ideal. We have taken the retrievals with an ideal ILS as
the reference. The fractional difference is defined here as,
$$D\% = \frac{X - X_{ref}}{X_{ref}} \times 100 \tag{11}$$

where $X$ is a vector which can include multiple elements such as gas profile or only
one element such as DOFs, root mean square of fitting residual (RMS), total column,
total random uncertainty, total systematic uncertainty, or total uncertainty. The total
random uncertainty and systematic uncertainty are the sum in quadrature of each
individual uncertainty listed in Table 2, and the total uncertainty is the sum in
quadrature of total random uncertainty and total systematic uncertainty. $X_{ref}$ is the
same as $X$ but for the nominal ideal ILS. For all gases, the retrievals with all levels of
ILS degradation fulfill the following filter criteria:
1) The RMSs of the residual (difference between measured and calculated spectra
after the fit) in all fitting windows has to be less than 3%.
2) The retrievals should converge for all levels of ILS degradation.
3) The concentrations of the target and interfering gases at each sub layer should be
positive.
4) The solar intensity variation (SIV) should be less than 10%. The SIV within the
duration of a spectrum is the ratio of the standard deviation to the average of the
measured solar intensities.
**5.1 ME amplitude and PE influence**
In order to determine how the ILS degradation affects the NDACC gas retrievals,



the results deduced from ILS considering both ME amplitude and PE are compared to
those only considering ME amplitude or PE. All types of ILS degradation in section 3
are used in this study. Fig.7 exemplifies the case of ILS *j*, where the differences in
total column, RMS, random uncertainty, systematic uncertainty, total uncertainty, and
DOFs for each gas relative to the retrieval with an ideal ILS are compared. Fig.8
shows the fractional difference in profile of each gas for ILS *j*. The results show that
the influence of ILS degradation on the total column, RMS, random uncertainty,
systematic uncertainty, total uncertainty, DOFs, and profile can be approximated by
the linear sum of individual ME amplitude influence and PE influence. The PE
influence is of secondary importance compared with the ME amplitude influence. The
comparisons for the results retrieved with ILS *a* to *i* come to the same conclusions.

Figs.9 and 10 show the influence of ILS *a* to *j* on total column and profile of all

NDACC gases. The resulting influence amounts depend on deviation amount and
deviation shape of ME. For positive MEs, in most cases, the ILS *j* causes the
maximum influence, and for negative MEs, the ILS *i* causes the maximum influence.
In a real instrument, the misalignment is a combination of misalignment *a* to *j*. In
principle, it should not cause influence exceeding misalignment *i* or *j* for the same
misalignment amplitude. In the following, misalignment *i* and *j* are selected on behalf
of negative and positive ME respectively to investigate how the ILS degradation
influence the NDACC gas retrievals.

## 5.2 Sensitivity study

We simulated seven levels of negative ME *i* and positive ME *j* with ALIGN60,

and incorporated them in the SFIT forward model, and then calculated the fractional
difference in various quantities for each gas relative to the retrieval with an ideal ILS.
The misalignments as inputs of ALIGN60 and the resulting ILSs are shown in Figs.
11 and 13. The corresponding Haidinger fringes at the maximum misalignment
position are shown in Figs. 12 and 14. The ME deviation, decenter of Haidinger
fringes and ILS deterioration varying over misalignment are evident. Fig.15 is the
sensitivity of total column with respect to different levels of ILS degradation. Figs. 16



~ 19 are the same as Fig. 15 but for DOFs, RMS, uncertainty and profile. The results
show that the ILS degradation affected total column, RMS, DOFs, retrieval
uncertainty, and profile. Generally, the larger the ME deviation, the larger the
influence. The positive and negative ME have opposite influence on total column,
DOFs, total uncertainty and profile.
With respect to total column, the influence of ILS degradation on stratospheric
gases is larger than the tropospheric gases. For $O_3$ and $HNO_3$, positive ME causes an
overestimated total column and negative ME causes an underestimated total column.
For other gases, negative ME causes an overestimated total column and positive ME
causes an underestimated total column. For all gases except $O_3$ and $CH_4$, the positive
ME influence is larger than the negative ME influence. For $O_3$ and $CH_4$, the negative
ME influence is larger than the positive ME influence.
For all gases, positive ME increases the DOFs and negative ME decreases DOFs.
For all gases except HF and $CH_4$, both positive ME and negative ME increase RMS.
For HF, positive ME increases RMS while negative ME decreases RMS. For $CH_4$,
positive ME decreases RMS and negative ME increases RMS.
The influence on systematic uncertainty and random uncertainty depends on ME
deviation type and gas type. The influence on total uncertainty is the combination of
the influence on total systematic uncertainty and total random uncertainty. For all
gases except $O_3$, positive ME decreases total uncertainty and negative ME increases
total uncertainty. For $O_3$, positive ME increases total uncertainty and negative ME
decreases total uncertainty.
The ILS degradation causes an evident difference in profile within the altitude
ranges that show high retrieval sensitivity in Fig.4. Positive ME has more influence on
tropospheric gas than negative ME. Whereas, negative ME has more influence on
stratospheric gas than positive ME.
**5.3 Discussion and analysis**
For each gas, the *a priori* covariance matrices of $\mathbf{S}_a$, $\mathbf{S}_\varepsilon$, and $\mathbf{S}_{var}$ are the same in
the aforementioned study. According to equations 6 ~ 8, we conclude that the ILS





degradation altered the weighting function matrix $\mathbf{K}_x$ and eventually altered the
quantities such as the total column, RMS, random uncertainty, systematic uncertainty,
total uncertainty, DOFs, and profile. The change of $\mathbf{K}_x$ is attributed to the fact that the
ILS degradation alters gas absorption line shape and hence alters the structure of
calculated spectra, and aggravates the mismatch between the calculated spectra and
the measured spectra.
The stratospheric gases are more sensitive to ILS degradation than the
tropospheric gases, and the $ClONO_2$ exhibits the largest sensitivity. This is because
the absorption structure in stratosphere is narrower than that in troposphere, and is
more easily affected by ILS degradation. We set the acceptable fractional difference in
total column for $ClONO_2$ and other NDACC gases as 10% and 1%, respectively.
Considering the excessively large of 28% ME deviation seldom occurred within
NDACC network because of the regular alignment at each site, the permitted
maximum ILS degradation for each gas is deduced in Table 5.

## 6 Consistency evaluation

For each gas, section 5 only selects one spectrum for study. This section uses all
spectra recorded at Hefei from September 2014 to April 2017 to evaluate the
consistency of above study. These spectra span a large difference in atmospheric water
vapor, SZAs, surface pressures, surface temperatures, wind speeds, and wind
directions (Fig. 20). All retrievals fulfill the above filter criteria are included in this
study. A simulated ILS $j$ with maximum ME amplitude deviation of 5% is used in the
retrieval. The results are compared to the retrievals deduced from an ideal ILS. The
Hefei site has run NDACC observations with the Bruker 125HR FTS for more than
two years. We regularly use a low-pressure HBr cell to diagnose the misalignment of
the spectrometer and to realign the instrument when indicated. For all spectra used in
this study, the ILS can be regarded as ideal and thus the retrievals with ideal ILS can
be taken as the reference.
Figs. 21 ~ 26 present the fractional difference in total column, RMS, total
uncertainty, and DOFs under different humidity, pressure, SZA, temperature, wind



direction, and wind speed. The results show that the fractional difference in total
column and total uncertainty for all gases are consistent under different
meteorological conditions and SZAs. The fractional difference in DOFs for all gases
except $N_2O$ and HCN are also consistent. For $N_2O$ and HCN, the variation of
fractional difference in DOFs is larger than that of total column and total uncertainty.
But most of them are less than 10% and independent of meteorological conditions and
SZAs. For most gases, the fractional difference in RMS exhibits more scatters than
the total column, total uncertainty, and DOFs. However, they are also independent of
meteorological conditions and SZAs, and most of them are less than 10%. In general,
the influence of ILS degradation on NDACC gases retrieval shows good consistency
under different meteorological conditions and SZAs.
**6 Conclusion**
We assessed the influence of instrumental line shape degradation on all current
NDACC gases retrieval via investigation of sensitivities of total column, root mean
square of fitting residual, total random uncertainty, total systematic uncertainty, total
uncertainty, degrees of freedom, and profile with respect to modulation efficiency
deviations. The study concluded that the influence of instrumental line shape
degradation can be approximated by the linear sum of individual modulation
efficiency amplitude influence and phase error influence. The phase error influence is
of secondary importance compared with the modulation efficiency amplitude
influence. The influence amounts depend on deviation amount and deviation shape of
the modulation efficiency.
For total column retrieval, the stratospheric gases are more sensitive to
instrumental line shape degradation than the tropospheric gases. For profile retrieval,
the positive modulation efficiency has more influence on tropospheric gases than the
stratospheric gases. While the negative modulation efficiency has more influence on
stratospheric gases than the tropospheric gases. The influence of instrumental line
shape degradation on NDACC gas retrievals shows good consistency under different
meteorological conditions and solar zenith angle. Finally, as summarized in Table 5,





we deduced maximum instrumental line shape deviations allowable for suppressing
the influence within a specified acceptable ranges.
**7 Acknowledgements**
This work is jointly supported by the National High Technology Research and
Development Program of China (No. 2016YFC0200800, 2016YFC0203302), the
National Science Foundation of China (No. 41605018, No. 41405134, No.41775025,
No. 41575021, No. 51778596, No. 91544212, No. 41722501), Anhui Province
Natural Science Foundation of China (No. 1608085MD79), and the German Federal
Ministry of Education and Research (BMBF) (Grant No. 01LG1214A). The
processing environment of SFIT4 and some plot programs are provided by National
Center for Atmospheric Research (NCAR), Boulder, Colorado, USA. The NDACC
networks are acknowledged for supplying the SFIT software and advice.

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

**9 Figs**





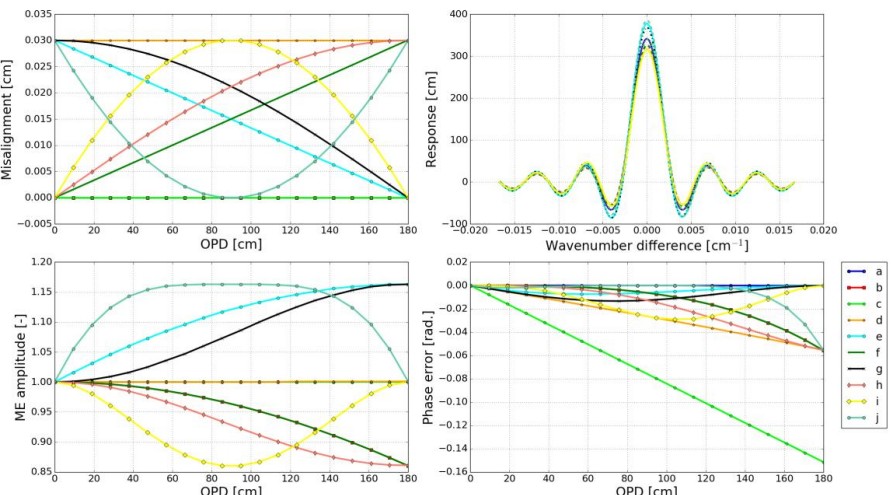

Fig.1. Simulated ILS degradation with respect to different types of misalignment. The results are
derived from ALIGN60. Top left demonstrates different types of misalignment (*a* to *j*) used in the
simulation, top right is the resulting ILS, bottom left is the resulting ME amplitude, and bottom
right is the resulting PE. Descriptions for the misalignment *a* to *j* are listed in Table 1.

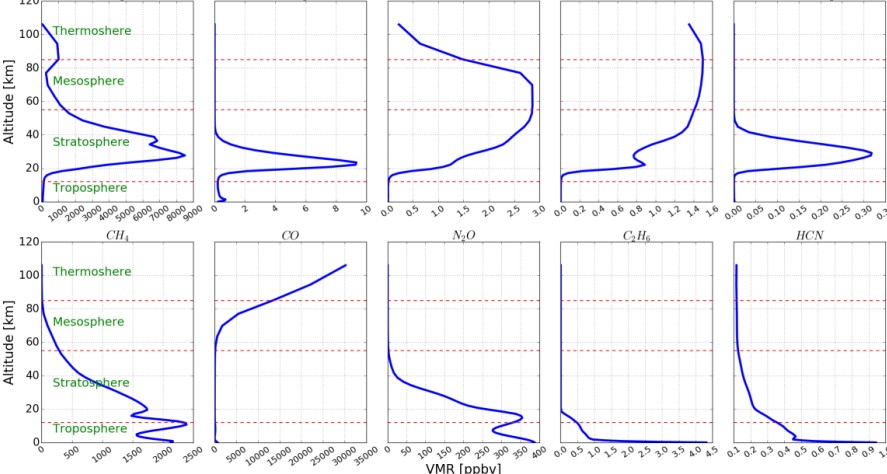

Fig.2. Typical profiles of ten NDACC gases. Bottom panels are five tropospheric gases, i.e., $CH_4$,
CO, $N_2O$, $C_2H_6$, and HCN. Top panels are five stratospheric gases, i.e., $O_3$, $HNO_3$, HCl, HF, and
$ClONO_2$. Although the CO concentration above 60 km is much higher than that in the troposphere,
it is regarded as tropospheric gas because it is an anthropologic pollution gas and shows large
variation in troposphere.





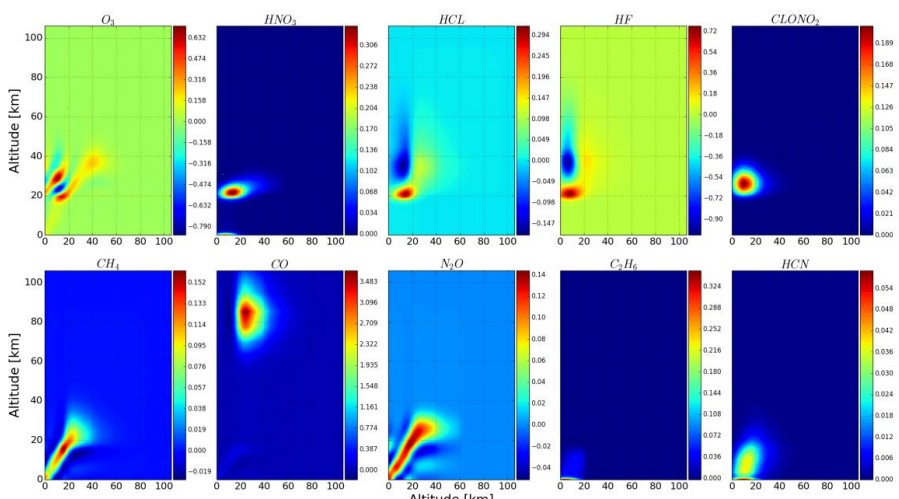

Fig.3. Averaging kernel matrices of ten NDACC gases. The deeper the color, the higher the
retrieval sensitivity. They are deduced from the spectra recorded at Hefei on September 8, 2015
with an ideal ILS.

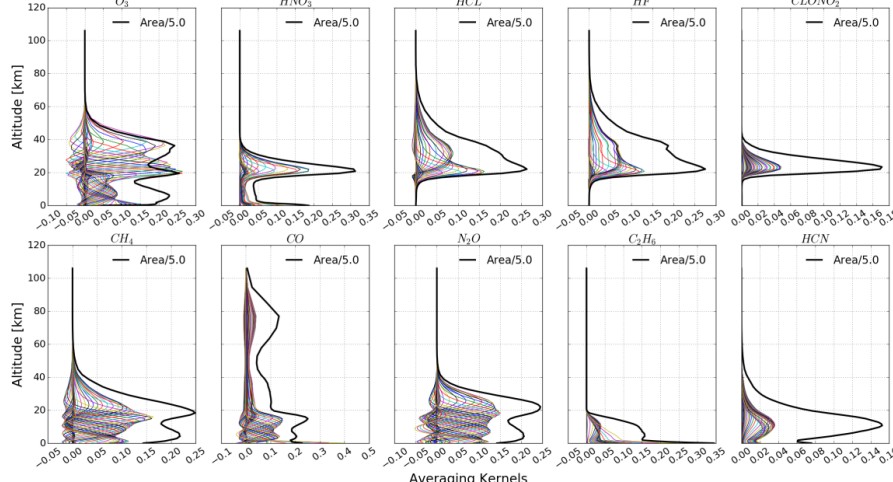

Fig.4 Averaging kernels of ten NDACC gases (color fine lines), and their area scaled by a factor of
0.2 (black bold line).





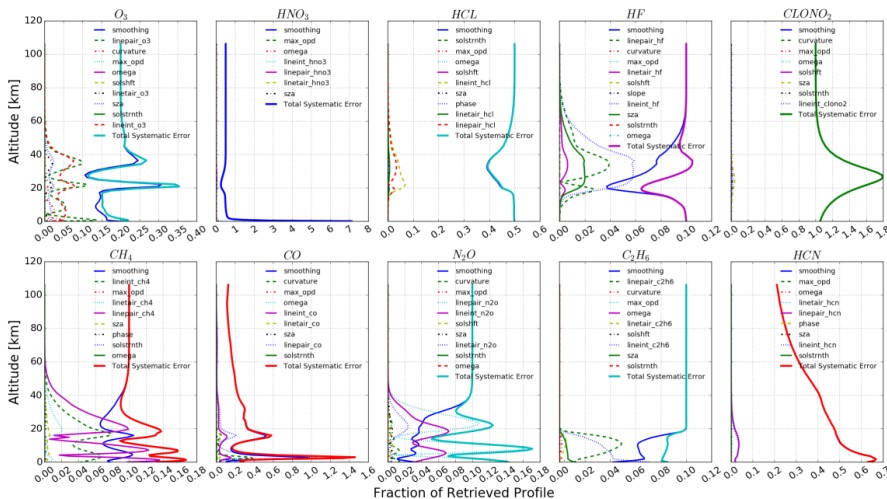


Fig.5. Ground-based FTIR systematic errors for ten NDACC gases retrieval. They are deduced
from the spectra used in section 5 with an ideal ILS.

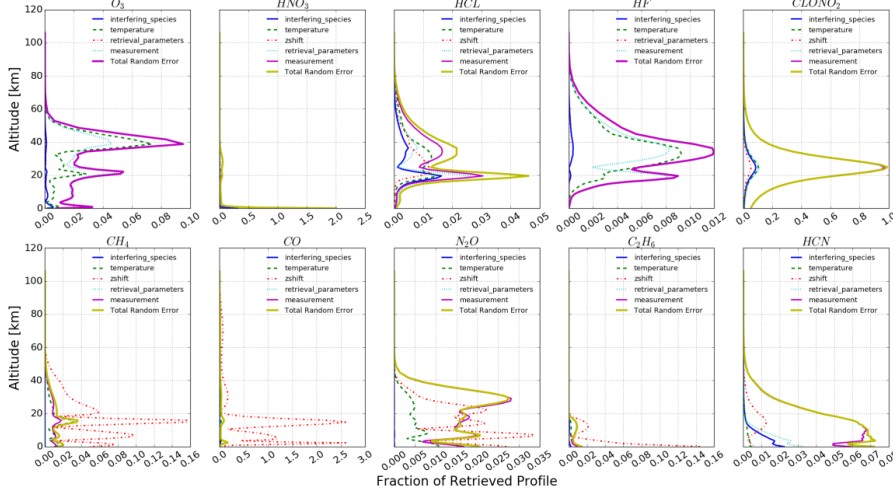


Fig.6. The same as Fig.5 but for random errors.



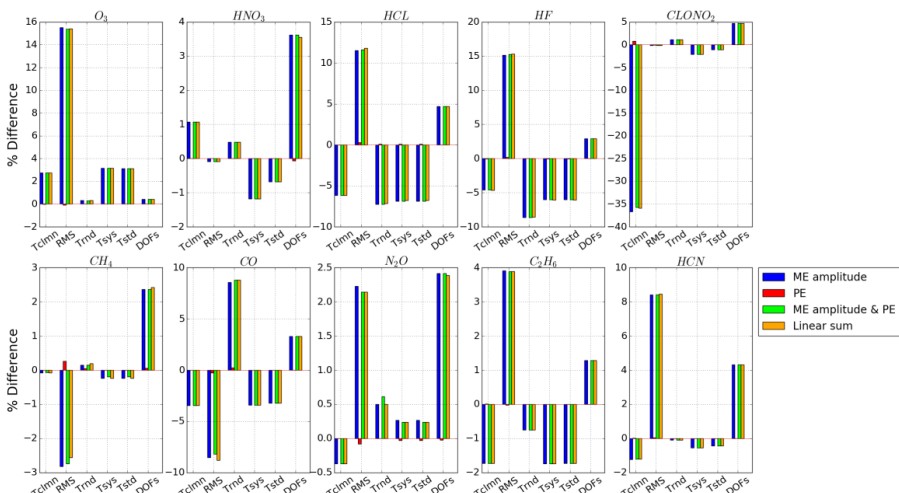

Fig.7. Fractional difference in total column, RMS, total random uncertainty, total systematic uncertainty, total uncertainty, and DOFs for misalignment *j*. "ME amplitude" represents the ILS only taken ME amplitude deviation into account. "PE" represents the ILS only taken PE deviation into account. "ME amplitude & PE" represents the ILS taken both ME amplitude and PE deviations into account. "Linear sum" represents the fractional difference of each item is linear sum of "ME amplitude" and "PE". The ME amplitude and PE are obtained from ALIGN60 with misalignment *j* in Fig.1.

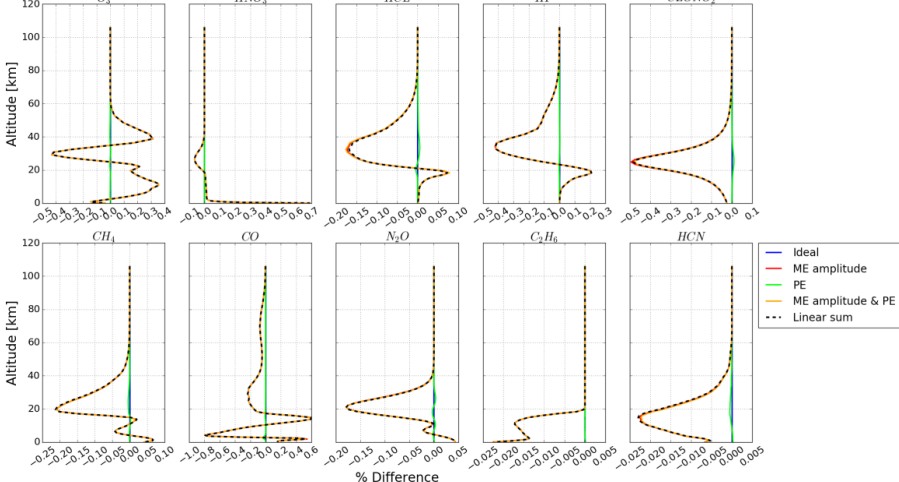

Fig.8. Fractional difference in profile for misalignment *j*. "ME amplitude" represents the ILS only taken ME amplitude deviation into account. "PE" represents the ILS only taken PE deviation into account. "ME amplitude & PE" represents the ILS taken both ME amplitude and PE deviations into account. "Linear sum" represents the fractional difference of each item is linear sum of "ME amplitude" and "PE". The ME amplitude and PE are obtained from ALIGN60 with misalignment *j* in Fig.1.





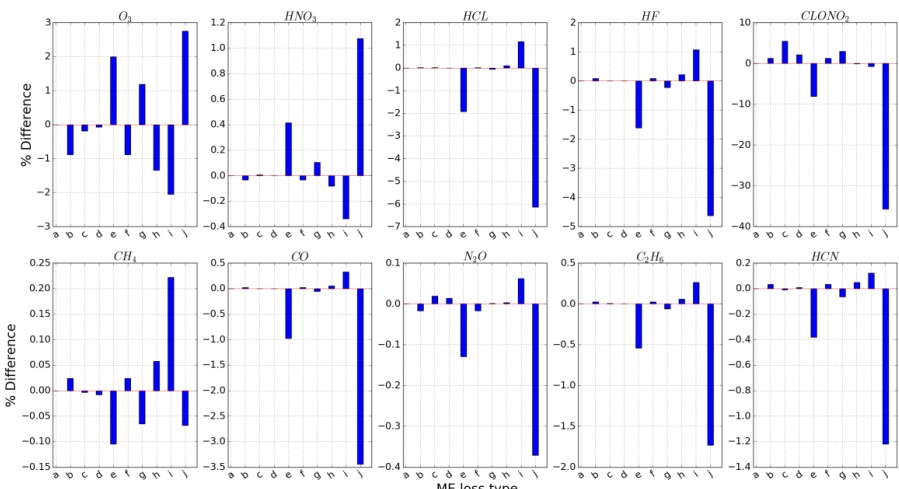


Fig.9. Sensitivity of total column to different types of ILS degradation. The ILS *a* to *j* correspond
to misalignment *a* to *j* in Table1.

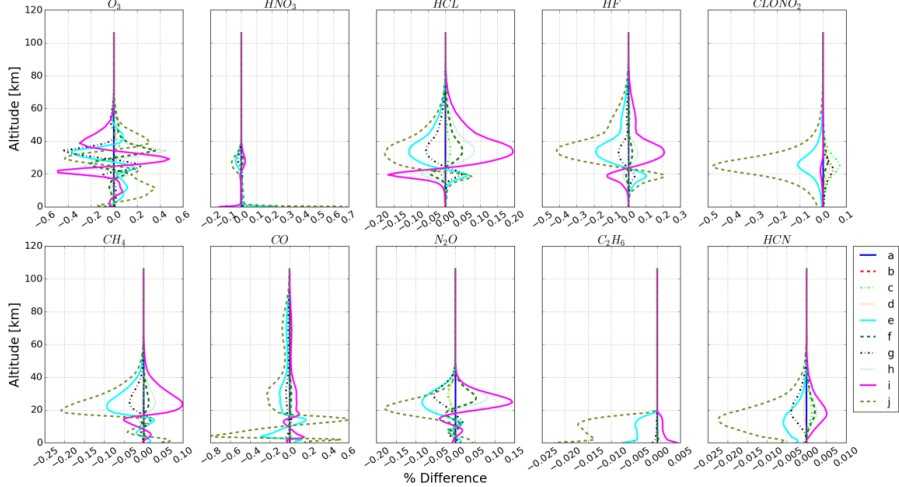


Fig.10. Sensitivity of profile to different types of ILS degradation. The ILS *a* to *j* correspond to
misalignment *a* to *j* in Table1.





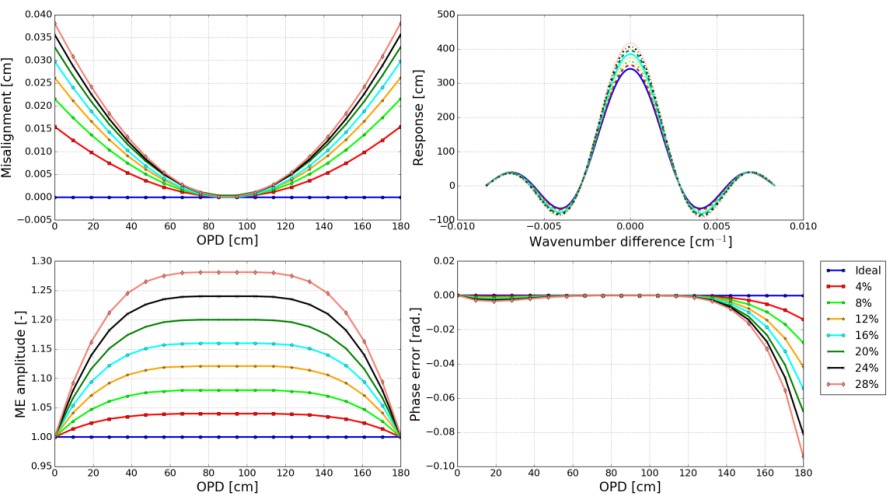


Fig.11. Simulated positive ME deviations along with OPD. Top left demonstrates the misalignment, top right is the resulting ILS, bottom left is the resulting ME amplitude, and bottom right is the resulting PE.


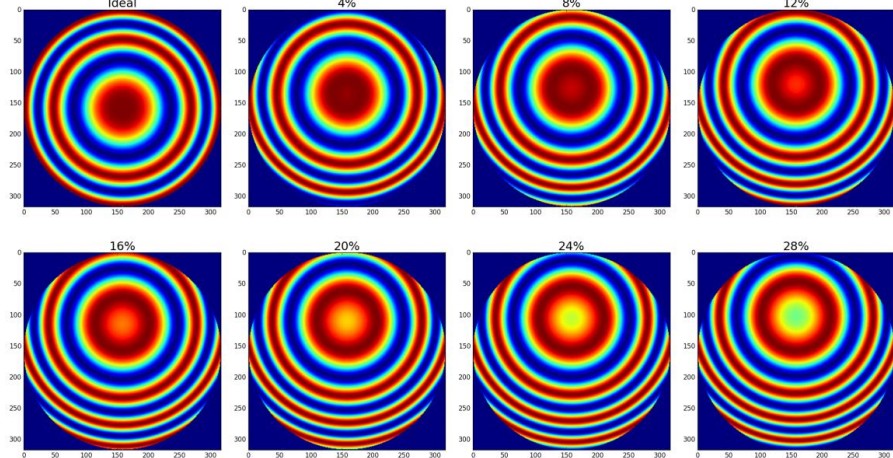


Fig.12. The Haidinger fringes at maximum OPD (the maximum misalignment position) for Fig. 11



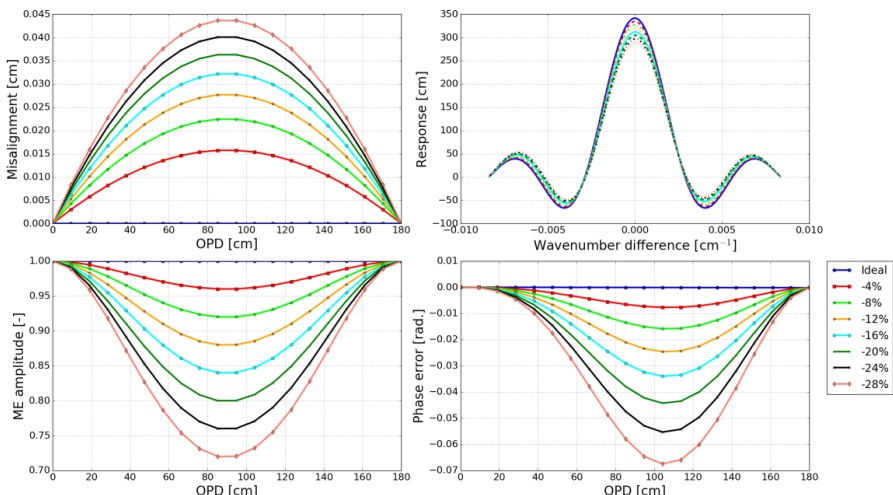


Fig.13. Simulated negative ME deviations along with OPD. Top left demonstrates the
misalignment, top right is the resulting ILS, bottom left is the resulting ME amplitude, and bottom
right is the resulting PE.

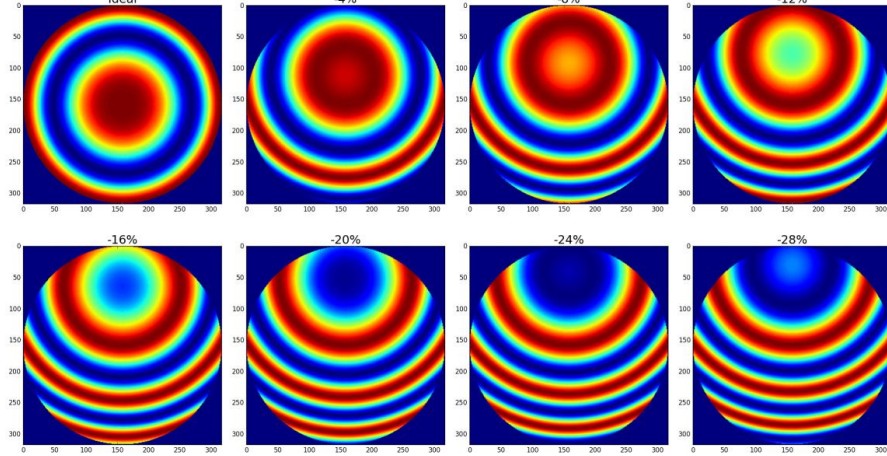


Fig.14. The Haidinger fringes at 1/2 maximum OPD (the maximum misalignment position) for Fig.

557    13






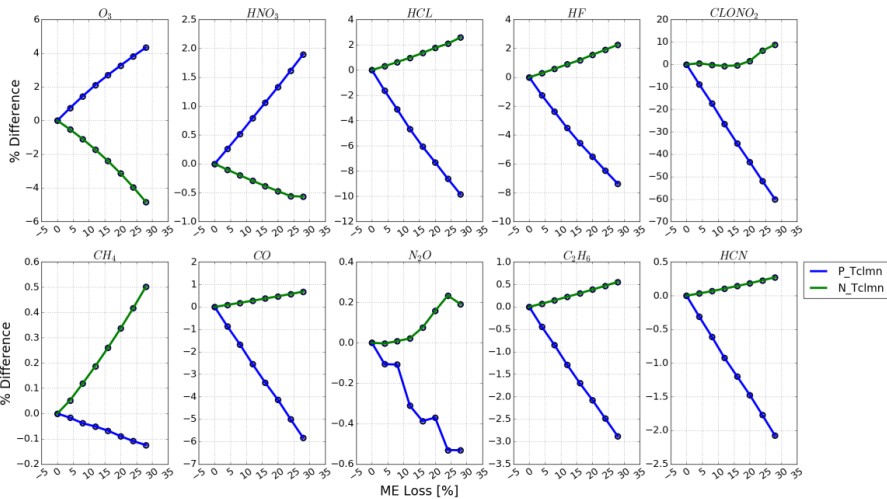


Fig.15. Sensitivity of total column with respect to ME deviation. "P_Tclmn" represents the
sensitivity of total column with respect to positive ME deviation and "N_Tclmn" represents the
sensitivity of total column with respect to negative ME deviation.

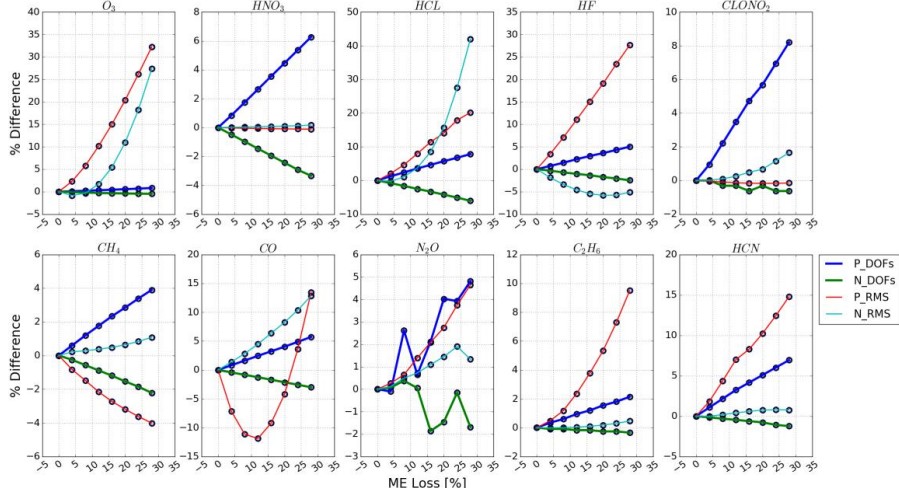


Fig.16. The same as Fig.15 but for DOFs and fitting RMS. The acronyms in the legend are similar
to those in Fig.15





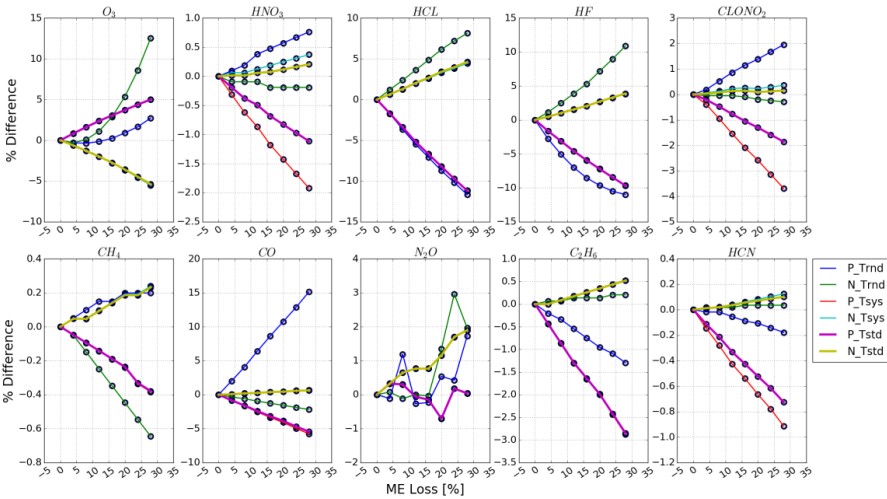


Fig.17. The same as Fig.15 but for total random uncertainty, total systematic uncertainty and total
uncertainty. The acronyms in the legend are similar to those in Fig.15. "Trnd", "Tsys" and "Tstd"
represent total random uncertainty, total systematic uncertainty and total uncertainty, respectively.

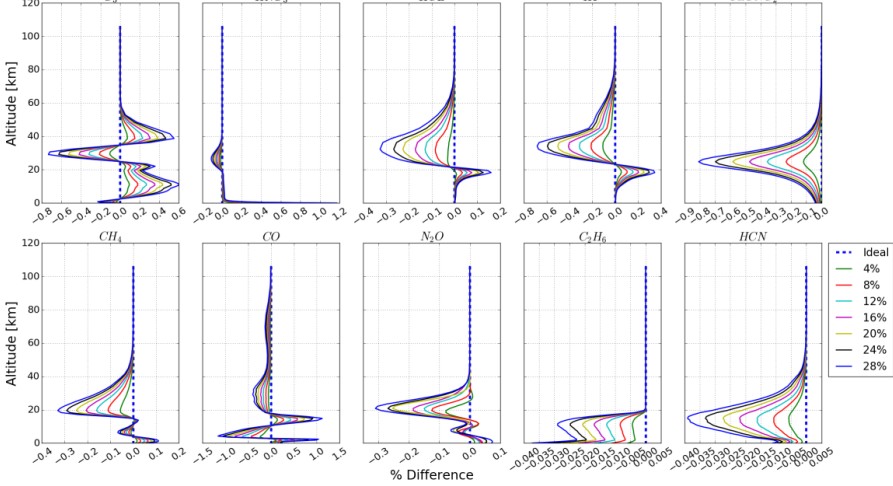


Fig.18. Sensitivity of profile with respect to ME deviation. "4%" represents the ME amplitude
deviation is 4%. The nomenclature for other plot labels is straightforward.





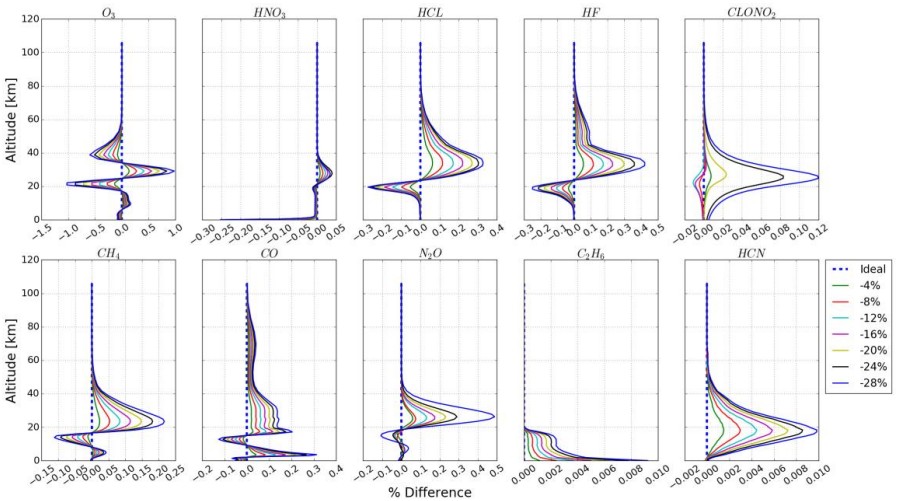


Fig.19. The same as Fig.18 but for negative ME deviation.

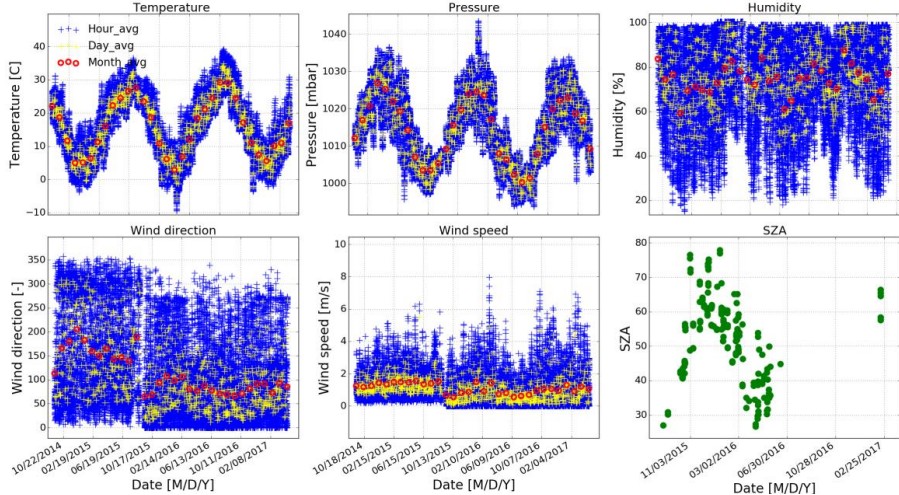


Fig.20. The meteorological data and SZAs record at Hefei from September 2014 to April 2017.
Large span of all these parameters are shown.

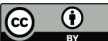



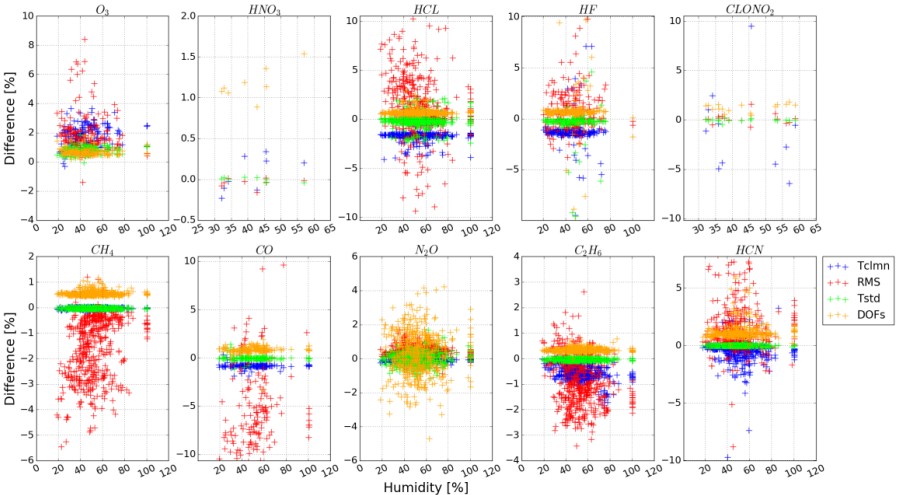

Fig.21. Fractional difference in total column, RMS, total uncertainty, and DOFs under different humidity conditions from September 2014 to April 2017 for ILS $j$ with a maximum ME deviation of 5%.

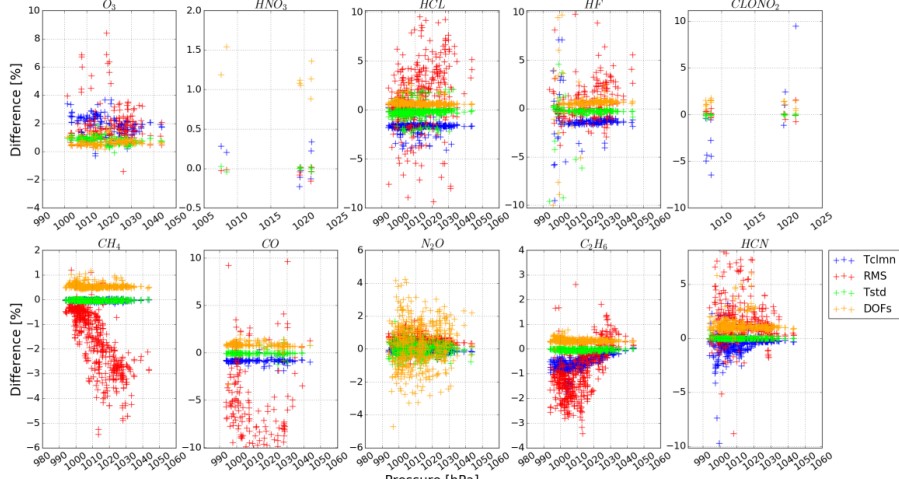

Fig.22. The same as Fig.21 but for surface pressure.





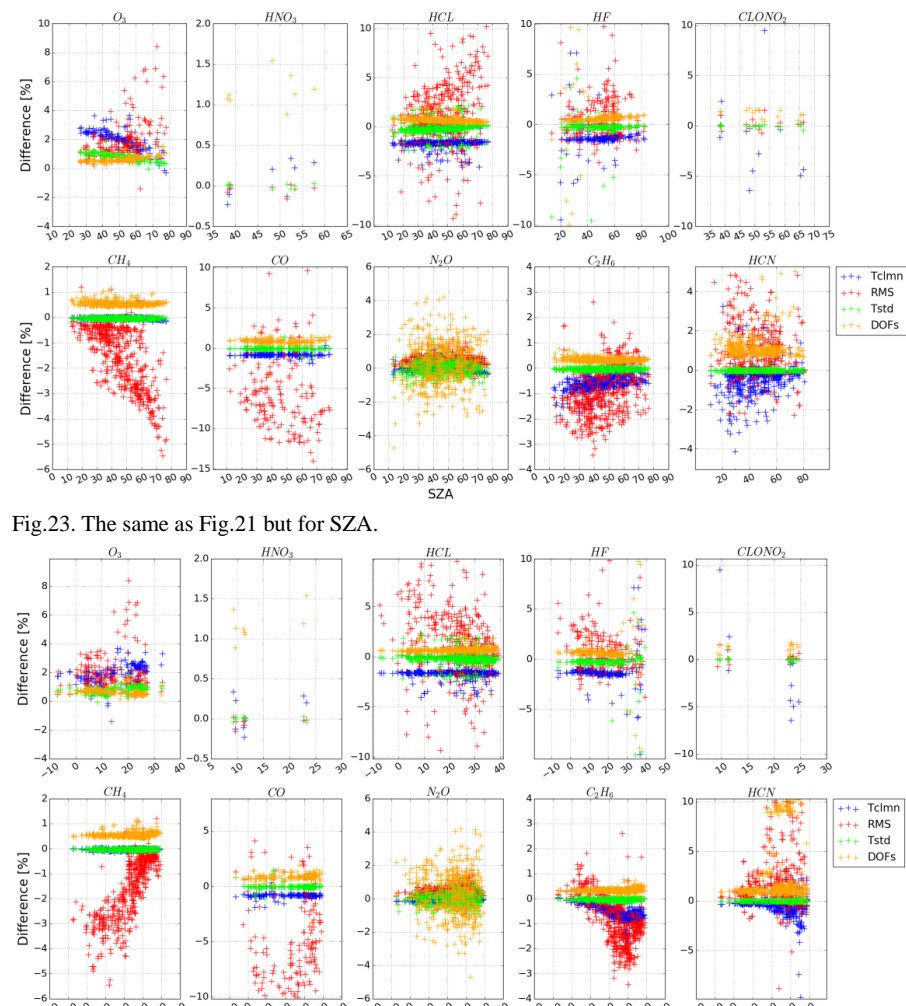


Fig.23. The same as Fig.21 but for SZA.

Fig.24. The same as Fig.21 but for surface temperature.



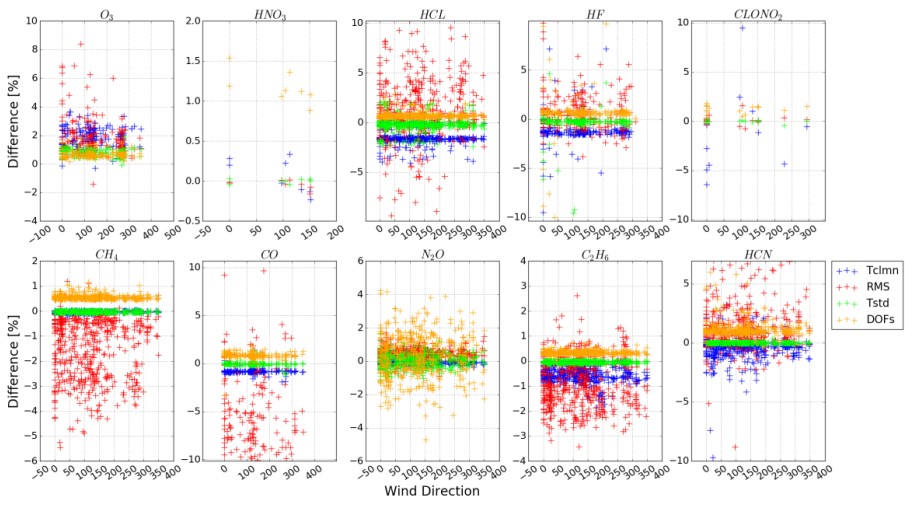

Fig.25. The same as Fig.21 but for wind direction.

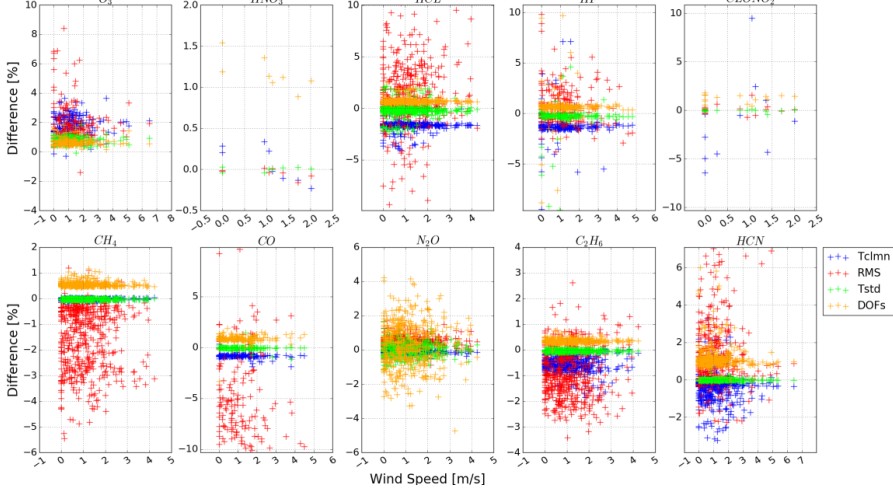

Fig.26. The same as Fig.21 but for wind speed





## 10 Tables

Table 1. Misalignments simulated in the ALIGN60

| Type [a] | Description | Input [b] | Output in maximum |
|---|---|---|---|
| a | No misalignment occurs: interferometer in ideal condition | none | ME amplitude: 1.00 PE: 0.000 rad. |
| b | Decenter of entrance field stop defining FOV: causes a linear increase in misalignment along OPD | 0.33 [mrad] field stop error | ME amplitude: 0.86 PE: -0.056rad. |
| c | Decenter of path measuring laser: causes a linear increase in phase error along OPD | 0.33 [mrad] laser error | ME amplitude:1.00 PE: -0.152rad. |
| d | Constant shear: causes a constant shear offset of fixed retro-reflector | 0.03 [cm] | ME amplitude: 1.00 PE: -0.056 rad. |
| e | Decreasing linear shear: causes a linear decrease in misalignment along OPD | 0.03-0.00017*OPD [cm] | ME amplitude: 1.16 PE: -0.007 rad. |
| f | Increasing linear shear: causes a linear increase in misalignment along OPD | 0.00017*OPD [cm] | ME amplitude: 0.86 PE: -0.056 rad. |
| g | Cosine bending of scanner bar: causes a cosine decrease in misalignment along OPD | $0.03*\cos(\pi*OPD/360)$ [cm] | ME amplitude: 1.16 PE: -0.013 rad. |
| h | Sine bending of scanner bar: causes a sine increase in misalignment along OPD | $0.03*\sin(\pi*OPD/360)$ [cm] | ME amplitude: 0.86 PE: -0.056 rad. |
| i | Cosine & sine bending of scanner bar: causes a chord increase in misalignment before 1/2 maximum OPD and causes a chord decrease in misalignment after 1/2 maximum OPD | $0.073*(\sin(\pi*OPD/360)+\cos(\pi*OPD/360))-0.073$ [cm] | ME amplitude: 0.86 PE: -0.029 rad. |
| j | Constant shear plus cosine & sine bending of scanner bar: causes a chordal decrease in misalignment before 1/2 maximum OPD and causes a chordal increase in misalignment after 1/2 maximum OPD | $-0.073*(\sin(\pi*OPD/360)+\cos(\pi*OPD/360))+0.103$ [cm] | ME amplitude: 1.16 PE: - 0.056 rad. |

[a] The b, f, h, and i are referred to increasing misalignment, the e, g, and j are referred to decreasing misalignment

[b] The input control file (i.e., align60.inp) for ideal condition is attached in the supplement. The input files for other gases can be straight forward.



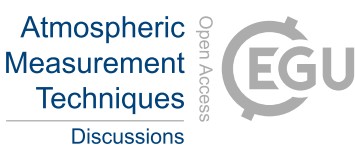

Table 2. Summary of the retrieval parameters used for all NDACC gases. All micro windows (MW) are given in cm$^{-1}$

| Gases | O$_3$ | HNO$_3$ | HCl | HF | ClONO$_2$ | CH$_4$ | CO | N$_2$O | C$_2$H$_6$ | HCN |
|---|---|---|---|---|---|---|---|---|---|---|
| Retrieval code | SFIT4 v 0.9.4.4 | SFIT4 v 0.9.4.4 | SFIT4 v 0.9.4.4 | SFIT4 v 0.9.4.4 | SFIT4 v 0.9.4.4 | SFIT4 v 0.9.4.4 | SFIT4 v 0.9.4.4 | SFIT4 v 0.9.4.4 | SFIT4 v 0.9.4.4 | SFIT4 v 0.9.4.4 |
| Spectroscopy | HITRAN2008 | HITRAN2008 | HITRAN2008 | HITRAN2008 | HITRAN2008 | HITRAN2008 | HITRAN2008 | HITRAN2008 | HITRAN2008 | HITRAN2008 |
| P, T profiles | NCEP | NCEP | NCEP | NCEP | NCEP | NCEP | NCEP | NCEP | NCEP | NCEP |
| A priori profiles for target/interfering gases except H$_2$O | WACCM | WACCM | WACCM | WACCM | WACCM | WACCM | WACCM | WACCM | WACCM | WACCM |
| MW for profile retrievals | 1000-1004.5 | 867.5-870 | 2727.73-2727.83 2775.7-2775.8 2925.8-2926.0 | 4109.4-4110.2 | 779.85-780.45 782.55-782.87 | 2613.7-2615.4 2835.5-2835.8 2921.0-2921.6 | 2057.7-2058 2069.56-2069.76 2157.5-2159.15 | 2441.8-2444.6 2481.2-2482.5 | 2976-2978 2982.6-2984.5 | 3268-3268.38 3287-3287.48 |
| Retrieved interfering gases | H$_2$O, CO$_2$, C$_2$H$_4$, O3668, O3686 | H$_2$O, OCS, NH$_3$ | CH$_4$, NO$_2$, O$_3$, N$_2$O, HDO | H$_2$O, HDO, CH$_4$ | O$_3$, HNO$_3$, H$_2$O, CO$_2$ | CO$_2$, NO$_2$, H$_2$O, HDO | O$_3$, N$_2$O, CO$_2$, OCS, H$_2$O | CO$_2$, CH$_4$ | H$_2$O, CH$_4$, O$_3$ | H$_2$O, O$_3$, C$_2$H$_2$, CH$_4$ |
| H$_2$O treatment — A priori profile | NCEP | NCEP | NCEP | NCEP | NCEP | NCEP | NCEP | NCEP | NCEP | NCEP |
| Fit in each WM | Profile retrieval | Scaling retrieval only | Profile retrieval | Profile retrieval | Scaling retrieval only | Profile retrieval | Profile retrieval | Profile retrieval | Profile retrieval | Profile retrieval |
| SNR for de-weighting | None | None | 300 | None | None | None | 500 | None | None | None |
| Regularization — S$_a$ | Diagonal: 20% No correlation | Diagonal: 50% No correlation | Diagonal: 50% No correlation | Diagonal: 10% No correlation | Diagonal: 100% Exponential correlation HWHM: 8 km | Diagonal: 10% No correlation | Diagonal: 11% ~ 27% No correlation | Diagonal: 10% No correlation | Diagonal: 10% No correlation | Diagonal: 21% ~ 79% No correlation |
| SNR | Real SNR | Real SNR | Real SNR | Real SNR | Real SNR | Real SNR | Real SNR | Real SNR | Real SNR | Real SNR |
| ILS | ALIGN60 | ALIGN60 | ALIGN60 | ALIGN60 | ALIGN60 | ALIGN60 | ALIGN60 | ALIGN60 | ALIGN60 | ALIGN60 |
| Error analysis | Systematic error: -Smoothing error (smoothing)[a] -Errors from parameters not retrieved by sfit4[b]: Background curvature (curvature), Optical path difference (max_opd), Field of view (omega), Solar line strength (solstrnth), Background slope (slope), Solar line shift (solshift), Phase (phase), Solar zenith angle(sza), Line temperature broadening (linetair_gas), Line pressure broadening (linepair_gas), Line intensity(lineint_gas) Random error: -Interference errors: retrieval parameters (retrieval_parameters), interfering species (interfering_species) -Measurement error (measurement) - Errors from parameters not retrieved by sfit4[b]: Temperature (temperature), Zero level (zshift) | | | | | | | | | |

[a] The bracket shows the corresponding acronym in Figs.5 and 6

[b] The input uncertainties of all these items are the same and are included into error analysis if they are not retrieved. Otherwise, the corresponding uncertainties wouldn't be included.



Table 3. Altitude ranges with sensitivity larger than 0.5 for all NDACC gases

| Items | $O_3$ | $HNO_3$ | HCl | HF | $ClONO_2$ | $CH_4$ | CO | $N_2O$ | $C_2H_6$ | HCN |
|---|---|---|---|---|---|---|---|---|---|---|
| Altitude ranges (km) | Ground - 44 | 17 - 28 | 18 - 42 | 18-44 | 20 - 28 | Ground - 31 | Ground - 27 | Ground - 31 | Ground - 13.5 | 4.5-18 |
| DOFs | 5.2 | 1.4 | 1.5 | 1.3 | 0.55 | 3.5 | 3.8 | 4.0 | 1.2 | 1.1 |





Table 4. The dominant sources of systematic errors and random errors for all NDACC gases

| Items | O₃ | HNO₃ | HCl | HF | ClONO₂ | CH₄ | CO | N₂O | C₂H₆ | HCN |
|---|---|---|---|---|---|---|---|---|---|---|
| Systematic error | smoothing error, line intensity, line pressure broadening | smoothing error | smoothing error | smoothing error, SZA line intensity, line pressure broadening | smoothing error | smoothing error, line pressure broadening, line intensity | smoothing error, line pressure broadening, curvature | smoothing error, line pressure broadening, line intensity | smoothing error, line intensity, line pressure broadening | smoothing error, line pressure broadening, line intensity |
| Random error | measurement error, temperature, | measurement error | Measurement error, retrieval parameters, interfering species, temperature | measurement error, temperature, | measurement error | zero level, measurement error, temperature | zero level | measurement error, zero level, temperature | zero level, measurement error | measurement error, interfering species, retrieval parameters, zero level |



Table 5. Recommendation for suppressing fractional difference in total column for ClONO$_2$ and other NDACC gases within 10% and 1%, respectively

| Items | O$_3$ | HNO$_3$ | HCl | HF | ClONO$_2$ | CH$_4$ | CO | N$_2$O | C$_2$H$_6$ | HCN |
|---|---|---|---|---|---|---|---|---|---|---|
| Positive ME | < 6% | <15% | <5% | <5% | <5% | * | <5% | * | < 9% | <13% |
| Negative ME | < 6% | * | <12% | <12% | * | * | * | * | * | * |

*The influence on ClONO$_2$ is less than 10% and on all other NDACC gases are less than 1% even the ILS degrade by an excessively large of 28%, and thus can normally be regarded as negligible.