# Peer review of "The influence of instrumental line shape degradation on NDACC gas retrievals: total column and profile"

_Atmospheric Measurement Techniques, 2017_

## Referee Comment (RC1) · Anonymous Referee #2 · 16 Feb 2018

This paper "The influence of instrumental line shape degradation on NDACC gas retrievals" by Sun et al presents sensitivity studies regarding the influence of ILS degradation in FTIR retrievals results. It is well known that the shape of the gas absorption lines can be impacted by the ILS, if instruments are not well-aligned. However, up to now there are not many published results regarding the quantitative impact. The lack of details in the impact of ILS makes this study important. The topic of the study is interesting and suitable for the journal. I suggest some revisions before its publication.

General Comments:

- The manuscript is short and lack important quantitative details regarding the finding with respect to the influence of ILS (results section). While the reader can check figures, and make sense of quantitative results authors do not explain in detail in the text

their findings (see specific comments below).

- Authors use an ideal ILS of actual FTIR measurements to know the influence of different ILS degradation. This statement is important since all quantitative Figures shown are with respect to this reference. However, there is a lack of proof about the ILS of actual measurements. If the ILS of actual measurements deviate from ideal the results shown here might change significantly. I suggest to include the actual ILS of the FTIR and its temporal variability.

- My understanding based on the analysis and table 5 is that the effect of the ILS (given that degradation of ILS for most FTIRs-NDACC is low) can be regarded as negligible for most gases, except, N2O, correct?. I do not find suggestions beside table 5 for including the ILS in the analysis of standard NDACC gases. Given that the ILS effect is negligible, would you suggest using ideal ILS?. I recommend to include a section with specific recommendations for FTIR/NDACC sites that will bring dialogue towards and harmonization in ILS.

- I recommend a thorough revision in format/style of the citations.

- I recommend a thorough English revision.

Specific Comments:

Abstract:

L27, I would change "current NDAC gases" with "current standard NDACC gases" since FTIR retrievals go beyond these mandatory gases.

L33-34, influence is written twice, remove one.

L38-40, "In order to suppress the influence on total column for ClONO2 and other NDACC gases within 10% and 1%, respectively, the permitted maximum ILS degradation for each NDACC gas was deduced (summarized in Table 5)". In my opinion, authors should summarize table 5 in the abstract rather than pointing the reader to

[Figure]

table 5. I found this difficult to interpret if the reader aims to check the abstract only.

Introduction:

L61, "FTIR spectrometers are highly precise and stable measurement devices and the instrumental line shapes (ILSs) not far from the theoretical limit if carefully aligned". This sentence is not clear. Please change it accordingly. Consider something like this: "FTIR spectrometers are highly precise and stable devices and if carefully aligned the instrumental line shape (ILS) might not be far from the theoretical limit".

L72-74. It might be important to mention that TCCON only uses NIR, fewer gases, and only columns are aimed compared to NDACC.

3 Simulation of ILS degradation

I could not find a description of ALIGN60 in the references provided. I suggest to describe in more detail ALIGN60 in this paper.

NDACC gases retrieval

4.1 Retrieval strategy

- L151-152. Is there a reference for the retrieval setting of NDACC?, if so cite it here. -The size of Table 2 can be significantly smaller. I suggest to remove all cells that are similar for the different gases and add a description in either the main text or caption of table, e.g., spectroscopy, P,T profiles, etc

4.2 Averaging kernels

- There are 26 Figures in the main text and I would consider removing some, e.g., Fig. 3 and 4 provide similar information. I would remove Fig 3 (or move it to supplemental information).

- Change to appropriate chemical formulas, e.g., HCL to HCl, etc.

4.3 Error Analysis

I would expect a description of the ILS in the uncertainty budget here. However, it is not clear how the error in ILS influences the uncertainty budget in either table 2 and Figs 5 and 6.

- In order to catch attention to the influence of ILS in the retrieval of gases I would remove Figs. 5 and 6. Again, I think 26 Figs are overwhelming. Instead, in table 4, which also does not add information, add quantitative numbers of leading/dominant errors including the ILS uncertainty.

5. ILS influence

- It is mention that the ILS degradation of the FTIR at Hefei is less than 1.3% but authors do not show how they infer this. This is key in order to avoid convolution problems with the different types of degradation.

- L243-247. It is not clear whether a single spectrum is used (what time, sza, conditions?) or all spectra recorded on Feb 16, 2017. Clarify.

- L255-265. Expand a description of the different filter criteria used here. It is clear that retrievals need to converge, but what about the 3% rms limit, what does SIV mean and why 10% is used?

- The color code of ME amplitude, PE, etc in Figs. 7 and 8 are different. To be consistent, change to same color code scheme. Remove the ideal case in Fig. 8.

- Do results shown in Figs 7 and 8 correspond to a single spectrum? if so include date/time in the caption.

- It is quite strange that % difference in total columns (Fig. 7) is larger than % Difference of profiles in Fig. 8. Maybe Fig.8 is only the fraction?

- Use appropriate name for gases, e.g., change HCL to HCl, etc.

5.1 ME amplitude and PE influence

- It is interesting to see in Fig. 7 that for some gases the % difference in RMS is negative, which would mean that the RMS of the reference is greater than using ILS degradation. Why would the rms be smaller using degraded ILS if the FTIR is characterized as ideal?

- In general, there is a lack of description in findings here. I recommend to have a more quantitative analysis and description of results in this section

---

## Referee Comment (RC2) · Anonymous Referee #1 · 22 Feb 2018

General comments:

The authors present a study on the influence of instrumental line shape (ILS) degradation on NDACC gas retrieval. Although this topic has been discussed in several NDACC infrared working group (IRWG) meetings in the past there is not so much in the literature, except for a few species such as ozone or water vapor. This paper describes this topic in detail for all the ten species which are mandatory to retrieve and for which a harmonized data analysis scheme is established within the IRWG.

Since it is well written and gives a comprehensive presentation of the influence of an imperfect ILS I recommend publishing this paper. This paper fits in the scope of AMT and will be useful for the IRWG.

[Figure]

Specific comments:

- Chapter 4.3 and Table 4: Channeling error is not included in the error analysis. At least for a weak absorber such as ClONO2 this error source is not negligible.

- While Haidinger fringes are presented for scenarios in Figs. 11 & 13 Haidinger fringes are missing for those in Fig. 1.

- The conclusion (as well as in the abstract) 'For total column retrieval, the stratospheric gases are more sensitive to instrumental line shape degradation than the tropospheric gases.' is a bit qualitative. I would suggest to add some numbers: For typical misalignment scenarios the column of O3, HCl, HF and ClONO2 changed by 3, 6, 5 and 35%, respectively.

- Table 5 nicely summarizes the recommendations for ME. I would suggest to add a sentence to the end of the abstract and the conclusion summarizing this result: 'For the retrieval of NDACC standard stratospheric species a ME within +-5% is required. Therefore, the alignment of an NDACC instrument needs to be better than 5% in terms of ME' or something similar.

Technical corrections:

- p. 5, line 139: increasing misalignment with increasing opd

- Legend of Figs. 5&6 and x axis description in Fig. 9 are hard to read (at least in my hardcopy).

- Figs.: 'l' in small letter in HCl and ClONO2

---

## Author Comment (AC1) · 2 Apr 2018

Response to Referee #2:

Thanks very much for your comments, suggestions and recommendation. Our response to all your comments are listed as follows. There is an extensive discussion among the authors regarding how to revise the content. So the response is delayed, and we are sorry for this.

This paper "The influence of instrumental line shape degradation on NDACC gas retrievals" by Sun et al presents sensitivity studies regarding the influence of ILS degradation in FTIR retrievals results. It is well known that the shape of the gas absorption lines can be impacted by the ILS, if instruments are not well-aligned. However, up to now there are not many published results regarding the quantitative impact. The lack of details in the impact of ILS makes this study important. The topic of the study is interesting and suitable for the journal. I suggest some revisions before its publication.

General Comments:

- The manuscript is short and lack important quantitative details regarding the finding with respect to the influence of ILS (results section). While the reader can check figures, and make sense of quantitative results authors do not explain in detail in the text their findings (see specific comments below).

**Response:** In the revised paper, we have added quantitative details regarding the finding of this study.

**Related change:** We have added quantitative details regarding the finding of this study in abstract, sections 6 and 7.

- Authors use an ideal ILS of actual FTIR measurements to know the influence of different ILS degradation. This statement is important since all quantitative Figures shown are with respect to this reference. However, there is a lack of proof about the ILS of actual measurements. If the ILS of actual measurements deviate from ideal the results shown here might change significantly. I suggest to include the actual ILS of the FTIR and its temporal variability.

**Response:** The Hefei site has run NDACC observations with the Bruker 125HR for more than three years. We regularly use a low-pressure HBr cell to diagnose the

misalignment of the spectrometer and to realign the instrument when indicated. As shown in Fig.5 (new added), all actual ILS degradations of the FTIR spectrometer within this selected period are less than 2% and can be regarded as ideal. The paper focuses on relative % difference of each quantity, the influence due to this assumption is negligible.

On the other hand, if the ILS of actual measurements deviate from ideal, it will cause an offset to both X and $X_{ref}$ in equation 9. Assuming the offset is $\Delta$, equation 9 becomes,

$$D = \frac{(X \pm \Delta) - (X_{ref} \pm \Delta)}{(X_{ref} \pm \Delta)} *100 = \frac{X - X_{ref}}{(X_{ref} \pm \Delta)} *100$$

Considering that $X_{ref}$ is close to ($X_{ref} \pm \Delta$), the results (especially the trends) shown here won't change significantly.

**Related change:** We have included the actual ILS of the FTIR and its temporal variability, i.e., Fig.5 in the revised version.

**-** (a) My understanding based on the analysis and table 5 is that the effect of the ILS (given that degradation of ILS for most FTIRs-NDACC is low) can be regarded as negligible for most gases, except, $N_2O$, correct?. (b) I do not find suggestions beside table 5 for including the ILS in the analysis of standard NDACC gases. (c) Given that the ILS effect is negligible, would you suggest using ideal ILS?. (d) I recommend to include a section with specific recommendations for FTIR/NDACC sites that will bring dialogue towards and harmonization in ILS.

**Response:**

(a) If total column is the target, the findings are the effect of the ILS degradation **cannot** be regarded as negligible for most gases, except, $N_2O$ and $CH_4$. Your understanding is up-side-down.

(b) In the revised paper, we have added quantitative details regarding the finding of this study. Please check abstract, sections 6 and 7 for details.

(c) If total column is the target, the ILS effect is not negligible for most gases, we suggest to keep the ILS degradation of each site within the recommendation. Note that the retrievals of certain gases, e.g., $O_3$, $CH_4$, $CO$, and $N_2O$, can be divided into

multiple independent sub layers depending on total DOFs. The recommendation don't apply to partial column integrated over each sub layer because, as Figs. 17 and 18 show, the sensitivity of profile to ILS degradation is altitude dependent. How ILS degradation influences partial column of each NDACC gas and how much ILS deviation from unity is acceptable if an ideal line shape is assumed beyond the scope of this paper and will be published elsewhere. Details can be found in section 6.

(d) In the revised paper, we have added quantitative details regarding the finding of this study. Please check abstract, sections 6 and 7 for details.

**Related change:** We have added quantitative details regarding the finding of this study in abstract, sections 6 and 7.

- I recommend a thorough revision in format/style of the citations.

**Response:** This has been done.

**Related change:** We have updated the format/style of the citations.

- I recommend a thorough English revision.

**Response:** The revised version is already gone through a copy-editing service.

**Related change:** A copy-editing service has been used.

Specific Comments:

Abstract:

L27, I would change "current NDACC gases" with "current standard NDACC gases" since FTIR retrievals go beyond these mandatory gases.

**Response:** This has been done in the revised version.

**Related change:** "current NDACC gases" becomes "current standard NDACC gases"

L33-34, influence is written twice, remove one.

**Response:** This has been done in the revised version.

**Related change:** We removed one of them

L38-40, "In order to suppress the influence on total column for ClONO2 and other NDACC gases within 10% and 1%, respectively, the permitted maximum ILS degradation for each NDACC gas was deduced (summarized in Table 5)". In my opinion, authors should summarize table 5 in the abstract rather than pointing the reader to table 5. I found this difficult to interpret if the reader aims to check the

abstract only.

**Response:** In the revised paper, we have added quantitative details regarding the finding of this study.

**Related change:** We have added quantitative details regarding the finding of this study in abstract, sections 6 and 7.

Introduction:

L61, "FTIR spectrometers are highly precise and stable measurement devices and the instrumental line shapes (ILSs) not far from the theoretical limit if carefully aligned". This sentence is not clear. Please change it accordingly. Consider something like this:

 "FTIR spectrometers are highly precise and stable devices and if carefully aligned the instrumental line shape (ILS) might not be far from the theoretical limit".

**Response:** We have revised this sentence as your suggestion.

**Related change:** We have revised this sentence as your suggestion.

L72-74. It might be important to mention that TCCON only uses NIR, fewer gases, and only columns are aimed compared to NDACC.

**Response:** This has been done as your suggestion.

**Related change:** A new sentence "The TCCON network only operates in near infrared (NIR) region and aims at column of fewer gases. While the NDACC network operates in both NIR and mid-infrared (MIR) regions and aims at both columns and profile of many gases." has been inserted in this section.

3 Simulation of ILS degradation

I could not find a description of ALIGN60 in the references provided. I suggest to describe in more detail ALIGN60 in this paper.

**Response:** A more detailed descriptions of ALIGN60 has been included.

**Related change:** A more detailed descriptions of ALIGN60 provided by the ALIGN60 developer has been included in this paper, please check section 3 in the revised version for details.

NDACC gases retrieval

4.1 Retrieval strategy

- L151-152. Is there a reference for the retrieval setting of NDACC?, if so cite it here.

**Response:** The retrieval setting of NDACC can be found via the link "([https://www2.acom.ucar.edu/irwg/links)](https://www2.acom.ucar.edu/irwg/links)".

**Related change:** We have cited the above link in the revised version.

-The size of Table 2 can be significantly smaller. I suggest to remove all cells that are similar for the different gases and add a description in either the main text or caption of table, e.g., spectroscopy, P,T profiles, etc

**Response:** All parameters that are the same for different gases are removed from Table 2, but the descriptions in the main text are kept. Now the size of Table 2 is significantly smaller than previous version.

**Related change:** We significantly shorten Table 2.

4.2 Averaging kernels

- There are 26 Figures in the main text and I would consider removing some, e.g., Fig. 3 and 4 provide similar information. I would remove Fig 3 (or move it to supplemental information).

**Response:** The previous Fig.3 is removed.

**Related change:** Previous Fig.3 is removed in the revised version.

- Change to appropriate chemical formulas, e.g., HCL to HCl, etc.

**Response:** This has been done.

**Related change:** All "HCL" and "CLONO$_2$" in this paper have been changed to "HCl" and "ClONO$_2$", respectively.

4.3 Error Analysis

I would expect a description of the ILS in the uncertainty budget here. However, it is not clear how the error in ILS influences the uncertainty budget in either table 2 or Figs 5 and 6.

**Response:** Detailed descriptions regarding how the error in ILS influences the quantities such as the total column, RMS, random uncertainty, systematic uncertainty, total uncertainty, DOFs, and profile as well as how much is acceptable can be found in the discussion, i.e., section 6. They are the purpose of this paper, should be present after the investigation (sensitivity study) rather than present before the investigation.

**Related change:** None

- (a) In order to catch attention to the influence of ILS in the retrieval of gases I would remove Figs. 5 and 6. (b) Again, I think 26 Figs are overwhelming. (c) Instead, in table 4, which also does not add information, add quantitative numbers of leading/dominant errors including the ILS uncertainty.

**Response:**

(a) Both figures have been removed in the revised version.

(b) Previous Figs.3, 5, and 6 have been removed, Figs.21- 26 tell the similar information and have been replaced by only one figure. However, as the referees' suggestions, we included one figure showing the actual ILS degradation and one figure showing Haidinger fringes of Fig.1. Now they are in total 20 figures.

(c) In order to catch attention to the influence of ILS in the retrieval of gases, we have significantly shorten the auxiliary contents. The error analysis may be important for study that focuses on retrieval itself, but contributes trivially to the main point of this paper. Table 4 has been removed in the revised version.

**Related change:** Figs.3, 5, and 6 have been removed, Figs.21- 26 have been replaced by one figure. One new figure showing the actual ILS degradation and one new figure showing Haidinger fringes of Fig.1 are added. Auxiliary contents, e.g., error analysis part have been shorten significantly and Table 4 has been removed.

5. ILS influence

- It is mention that the ILS degradation of the FTIR at Hefei is less than 2% but authors do not show how they infer this. This is key in order to avoid convolution problems with the different types of degradation.

**Response:** The Hefei site has run NDACC observations with the Bruker 125HR for more than three years. We regularly use a low-pressure HBr cell to diagnose the misalignment of the spectrometer and to realign the instrument when indicated. As shown in Fig.5, all actual ILS degradations of the FTIR spectrometer within this selected period are less than 2% and can be regarded as ideal. The paper focuses on relative % difference of each quantity, the influence due to this assumption is negligible.

**Related change:** We have included the actual ILS of the FTIR and its temporal

variability, i.e., Fig.5 in the revised version.

- L243-247. It is not clear whether a single spectrum is used (what time, sza, conditions?) or all spectra recorded on Feb 16, 2016. Clarify.

**Response:** Unlike TCCON network, the NDACC has seven consecutive optical filters to reduce the broadband signal (avoiding detector non-linearity) and it is not possible to retrieve all ten mandatory species within one filter spectra. For your comments, our answers are: the statistical analysis of each gas is based on a single spectrum, but different gas may use different filter spectrum. Thus, 5 spectra in total are used here but all of them are recorded on Feb. 16, 2016.

**Related change:** A statement has been include in section 5.

- L255-265. Expand a description of the different filter criteria used here. It is clear that retrievals need to converge, but what about the 3% rms limit, what does SIV mean and why 10% is used?

**Response:** These criteria are used to remove those spectra that have sampling errors or contaminated by aerosols, clouds, hazes or other unpredictable objects which cause a low SNR or a large detecting intensity variation. These spectra normally show bad fitting RMSs or in accuracy retrievals. We have included this clarification in the revised paper.

**Related change:** An expand description has been included here.

- The color code of ME amplitude, PE, etc in Figs. 7 and 8 are different. To be consistent, change to same color code scheme. Remove the ideal case in Fig. 8.

**Response:** These problems have been solved and now the two figures are consistent.

**Related change:** We updated the two figures, please check Figs.5 and 6 in the revised version for details.

- Do results shown in Figs 7 and 8 correspond to a single spectrum? if so include date/time in the caption.

**Response:** The NDACC has seven consecutive optical filters to reduce the broadband signal (avoiding detector non-linearity) and it is not possible to retrieve all ten mandatory species within one filter spectra. The statistical analysis of each gas is based on a single spectrum, but different gas may use different spectrum. Thus, 5

spectra in total are used here but all of them are recorded on Feb. 16, 2016.

**Related change:** A statement has been include in section 5.

- It is quite strange that % difference in total columns (Fig. 7) is larger than % Difference of profiles in Fig. 8. Maybe Fig.8 is only the fraction?

**Response:** After a careful check to our python scripts, we found we forgot to include a factor of 100 (see equation (9)) in calculations of fractional difference in profile, but all other quantities don't have this problem. Yes, all profile related figures in previous version do indicate the fraction rather than % difference. However, in the revised version, a factor of 100 has been included and the problems in all profile related figures have been solved. Thanks very much for pointing out these problems.

**Related change:** All profile related figures have been multiplied by a factor of 100.

- Use appropriate name for gases, e.g., change HCL to HCl, etc.

**Response:** This has been done.

**Related change:** All "HCL" and "CLONO$_2$" in this paper have been changed to "HCl" and "ClONO$_2$", respectively.

5.1 ME amplitude and PE influence

- It is interesting to see in Fig. 7 that for some gases the % difference in RMS is negative, which would mean that the RMS of the reference is greater than using ILS degradation. Why would the rms be smaller using degraded ILS if the FTIR is characterized as ideal?

**Response:** This "abnormal" phenomenon also troubled us for quite a long time during proceeding this study. Finally, we were lucky to figure out the reasons. This is because, for certain cases, the number of iterative step (to get converge) is different when using different ILS. For an example, the CO retrieval may converge with 3 iterative steps with the ideal ILS, but may need 4 or 5 iterative steps to get converge with degraded ILS. Thus, it is possible that the RMS of the reference is greater than using ILS degradation. This phenomenon do not occur quite often but indeed exist.

**Related change:** None

- In general, there is a lack of description in findings here. I recommend to have a more quantitative analysis and description of results in this section

**Response:** In the revised paper, we have added quantitative details regarding the finding of this study.

**Related change:** We have added quantitative details regarding the finding of this study in abstract, sections 6 and 7.

---

## Author Comment (AC2) · 2 Apr 2018

Response to Referee #1:

Thanks very much for your comments, suggestions and recommendation with respect to publish this paper in AMT. Our response to all your comments are listed as follows. There is an extensive discussion among the authors regarding how to revise the content. So the response is delayed, and we are sorry for this.

General comments:

The authors present a study on the influence of instrumental line shape (ILS) degradation on NDACC gas retrieval. Although this topic has been discussed in several NDACC infrared working group (IRWG) meetings in the past there is not so much in the literature, except for a few species such as ozone or water vapor. This paper describes this topic in detail for all the ten species which are mandatory to retrieve and for which a harmonized data analysis scheme is established within the IRWG. Since it is well written and gives a comprehensive presentation of the influence of an imperfect ILS I recommend publishing this paper. This paper fits in the scope of AMT and will be useful for the IRWG.

Specific comments:

- Chapter 4.3 and Table 4: Channeling error is not included in the error analysis. At least for a weak absorber such as ClONO2 this error source is not negligible.

**Response:** The selection of error items and their values cannot be easily standardised because most of them are instrument/site dependent. In this paper, we already included most common error items in the error analysis. The channeling error was not included because: 1, it is instrument dependent and it is not a common error, some instrument may have very weak channeling effect; 2, The main point of this paper is the same regardless of including or not including channeling error. This is because error analysis is the post processing (last step) of NDACC retrieval, how many errors to be included may have influence on the total error and thus the fractional difference of statistic errors, but have no influence on the total column, DOFs and profile which are obtained before post processing step.

**Related change:** None

- While Haidinger fringes are presented for scenarios in Figs. 11 & 13 Haidinger

fringes are missing for those in Fig. 1.

**Response:** We have included Haidinger fringes for those in Fig.1.

**Related change:** In the revised paper, Fig.2 showing Haidinger fringes for those in Fig.1 is included.

- The conclusion (as well as in the abstract) 'For total column retrieval, the stratospheric gases are more sensitive to instrumental line shape degradation than the tropospheric gases.' is a bit qualitative. I would suggest to add some numbers: For typical misalignment scenarios the column of O3, HCl, HF and ClONO2 changed by 3, 6, 5 and 35%, respectively.

**Response:** We have improved this description as your suggestion.

**Related change:** A quantitative description "For a typical ILS degradation (10%), the total columns of stratospheric gases $O_3$, $HNO_3$, HCl, HF, and $ClONO_2$ changed by 1.9%, 0.7%, 4%, 3%, and 23%, respectively. While the columns of tropospheric gases $CH_4$, CO, $N_2O$, $C_2H_6$, and HCN changed by 0.04%, 2.1%, 0.2%, 1.1%, and 0.75%, respectively." have been included in both abstract and conclusion. Please check the conclusion and abstract sections for details.

- Table 5 nicely summarizes the recommendations for ME. I would suggest to add a sentence to the end of the abstract and the conclusion summarizing this result: 'For the retrieval of NDACC standard stratospheric species a ME within +-5% is required. Therefore, the alignment of an NDACC instrument needs to be better than 5% in terms of ME' or something similar.

**Response:** We have added some sentences to summarize Table 4 (i.e., Table 5 in the previous version).

**Related change:** Some sentences to summarize Table 4 (Table 5 in the previous version) are added, please check section 6 for details.

Technical corrections:

- p. 5, line 139: increasing misalignment with increasing opd

**Response:** We have revised this sentence as your suggestion.

**Related change:** Now it is "Typically, the increasing misalignment with increasing OPD (b, f, h or i) causes negative ME amplitude and the decreasing misalignment

with increasing OPD (e, g or j) causes positive ME amplitude."

- Legend of Figs. 5&6 and x axis description in Fig. 9 are hard to read (at least in my hardcopy).

**Response:** In order to make the content more concise on the main point of this paper and catch attention to the influence of ILS on the retrieval of gases, following referee #2' suggestion, we have removed Figs. 5&6 which contribute trivial to the main point of this paper. We have updated Fig.9, now x axis description is very clear.

**Related change:** We removed Figs. 5&6 and updated the x axis of Fig.9.

- Figs.: 'l' in small letter in HCl and ClONO2

**Response:** All "HCL" and "CLONO$_2$" in this paper have been changed to "HCl" and "ClONO$_2$", respectively.

**Related change:** We have revised all "HCL" and "CLONO$_2$" as "HCl" and "ClONO$_2$", respectively.